# The Portofino Conglomerate (Eastern Liguria, Northern Italy): Provenance, Age and Geodynamic Implications

Federico Mantovani [1], Franco Marco Elter [1,*], Enrico Pandeli [2], Antonino Briguglio [1] and Michele Piazza [1]

1 Department of Earth, Environment and Life Science (DISTAV), University of Genova, 16132 Genova, Italy; federico.mantovani@edu.unige.it (F.M.)

2 Earth Science Department (DST), Geosciences and Earth Resources Institute (CNR-IGG), National Research Council, University of Florence, 50121 Florence, Italy

* Correspondence: franco.elter@unige.it

**Abstract:** The Portofino Conglomerate (PC) cropping out in the Eastern Liguria is an approximately 500 m thick, very gently folded succession mainly composed of poorly bedded and mostly matrix-supported conglomerates. It stratigraphically rests on the Helminthoid Flysch (UA3) thrusted onto the Antola Unit. We vertically distinguished three mostly ruditic litho/petrofacies: (i) Paraggi (fP) with carbonate clasts from an Helminthoid Flysch succession; (ii) Monte Pallone (fMP) with prevailing carbonate and meta-carbonate clasts and minor quartz-rich (meta)siliciclastic and high-pressure–low-pressure (HP-LP) metabasite clasts; and (iii) Monte Bocche (fMB) with dominant quartz-rich (meta)siliciclastic, meta-carbonate clasts, and minor granitoid elements and medium-temperature–high-temperature (MT-HT) regional metamorphic rocks. The middle-upper Eocen age of Paraggi litho/petrofacies is constrained by well-preserved microforaminifers (e.g., *Globigerinatheka*) recovered in the matrix. During its sedimentation, the directions of the paleocurrents would indicate that the PC underwent a counterclockwise rotation coeval with the first Cenozoic rotational phase of the Sardinia–Corsica system (50–30 Ma) and then stopped before the sedimentation of the Monte Pallone and Monte Bocche litho/petrofacies. The vertical compositional variation in the sedimentary inputs suggested that the PC is the result of a progressive deepening of the erosional level of a tectonic pile that can be located in the Ligurian Alps Chain. We considered the PC as the likely apical part of a submarine fan deposited in a piggy-back/thrust-top basin within the Alpine nappe stack. This sedimentary body was later tectonically transported eastward with its UA3 Helminthoid Flysch substrate (similarly to Epiligurian Units of the Northern Apennines) onto the Apenninic orogenic system (i.e., the Antola Unit).

**Keywords:** Portofino Conglomerate; Ligurian Alps; Northern Apennine; Eocene





## 1. Introduction

Within the complex puzzle of the Alpine and Northern Apennines syn/late- and post-orogenic basins, the Portofino Conglomerate (PC) is one of the less studied depositional units [1–4].

The geology framework of Liguria (northwestern Italy; Figure 1) is characterized by different paleogeographic domains [5]; from west to east are: the Provençal Domain, (characterized by a Paleozoic? Crystalline basement and its Carboniferous to Oligocene cover), Briançonnais Domain (characterized by a Paleozoic? Crystalline basement with its upper Carboniferous–Cenozoic cover), Piedmontese Domain (characterized by Meso-Cenozoic cover), and the Piedmontese-Ligurian and Ligurian Domains (characterized by different oceanic successions, part of which show polyphase Alpine tectono-metamorphic evolutions).

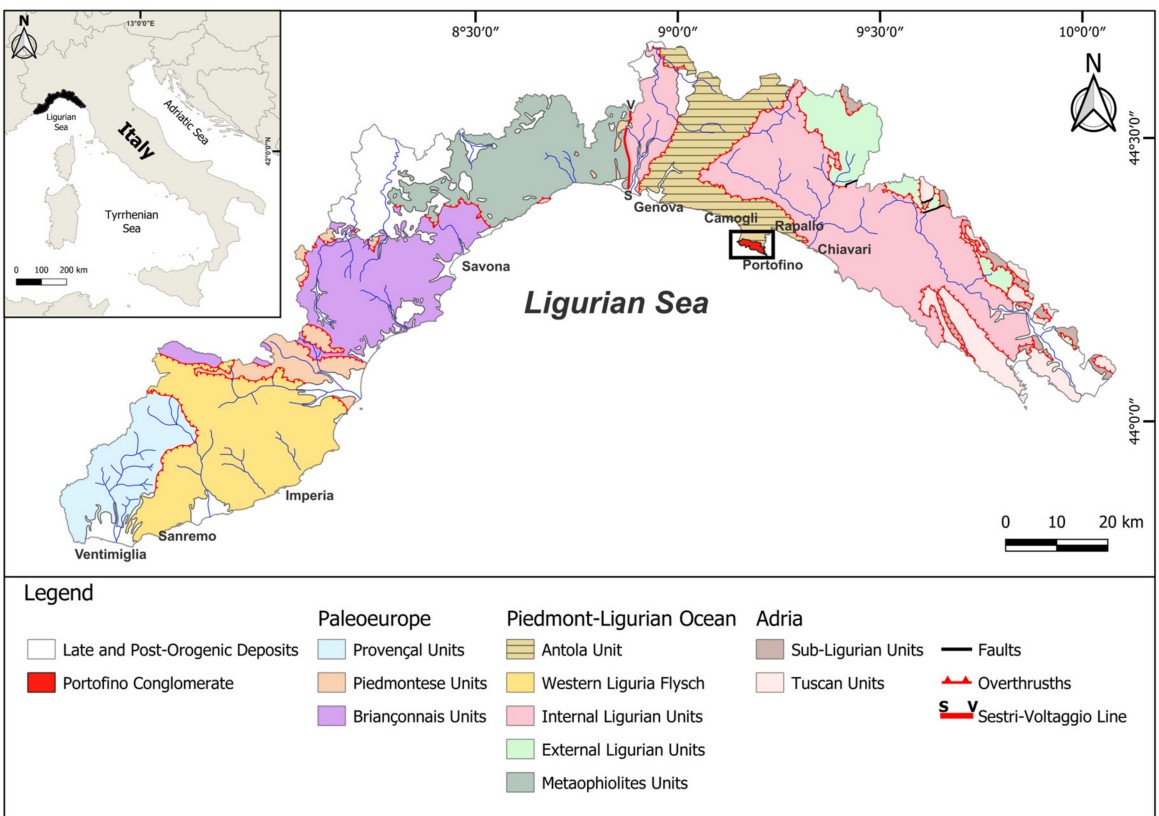

**Figure 1.** Simplified geological map of Liguria (modified after [6]). Black rectangle: studied area.

The PC crops out in the eastern Liguria (Riviera di Levante, northwestern Italy) and forms the homonymous promontory between Camogli to the northwest and Rapallo to the northeast (Figure 2).

The PC is a more than 500 m thick succession that stratigraphically rests on a tectonic pile of Helminthoid Flysch Units (i.e., the Antola Unit Auct. [7,8]) and is mainly composed of poorly bedded conglomerates with few sandstone and siltstone interbedded. The nature of the clasts [2,4] pointed out the presence of HP-LP metabasite clasts. The paleoenvironment and age of the PC are still not well constrained (lower Oligocene marine deposit in [1]) because of the rarity of fossils. Moreover, the definition of the relationship between the PC and Antola Unit s.l. recalls that between the late- to post-orogenic succession of the Tertiary Piedmont Basin (TPB; late Eocene—early Oligocene in [9,10]) and the underlying Antola Unit [11–14] in the outer northwestern part of the Apennine chain, it was important at a regional scale for the reconstruction of the Neogene geodynamics.

The aim of this paper was to identify and describe in detail the stratigraphic–sedimentologic and petrographic features of the three main litho/petrofacies within the Portofino Conglomerate clastic succession. Finally, the presented data allowed us to discuss the age, source area, and paleogeographic location of the PC depositional system and its Ligurian substrate. All of this will improve the geodynamic reconstruction of the Ligurian Alps and the Northern Apennine Chain systems.

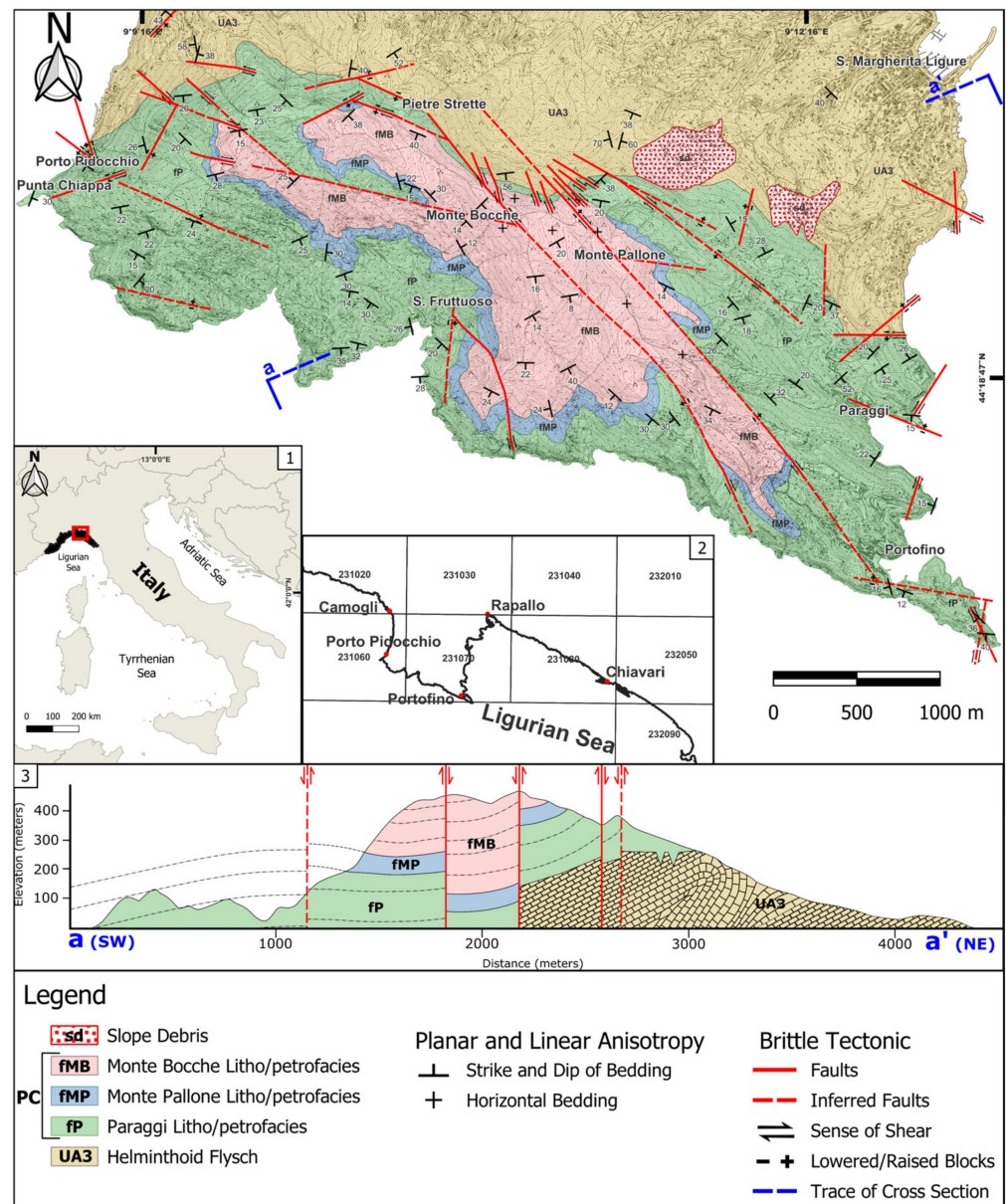

**Figure 2.** Geological map of the Portofino Promontory: (1) geographic insert showing the position of Liguria (in black) and the studied area (red square); (2) geographic insert showing the locality in the surrounding area of the Portofino Promontory and relative sheet number of the 1:10,000 geological maps published in the literature; (3) schematic geological cross section.

## 2. Materials and Methods

The authors carried out a 1:10,000-scale geological survey as well as stratigraphic–sedimentologic and petrographic studies on the different outcrops of the Portofino Conglomerate between Santa Margherita Ligure and Rapallo and analyzed its relationships with the underlying Helminthoid Flysch Units.

To characterize the litho/petrofacies of the PC together with the lithological/sedimentological analyses of the outcrops (including paleocurrent measurements through the imbrication of the clasts), several square meshes of one meter per side were defined vertically in each of the three sedimentary units to cover their whole thickness; 85 meshes in total were obtained in stratigraphic order within the PC. Inside each mesh, more than 500 clasts were counted according to the lithology for defining (in percentage) any vertical compositional

variations in the clastic assemblages. For each clast, we also defined its shape, roundness, and sphericity.

More than 300 imbrication planes were taken and plotted in "Stereonet 11" software v. 11.5.1 (equal area lower hemisphere) for the calculation of the poles and relative contours.

The geological maps were produced with "QGis™" software v. 3.22.11, while the geological section was sketched with "Inkscape: Open Source Scalable Vector Graphics Editor™" v. 1.2.2. Three-dimensional models and representations were created with "Vectary™" software v. 3.0.

## 3. Geological Framework of the Studied Area

The geological framework of the Portofino Promontory consists in the stratigraphic superposition of the PC onto the Cretaceous–early Cenozoic Ligurian Helminthoid Flysch succession; in the past all of which have been attributed to the Antola Flysch Auct [15,16].

The Antola Flysch or Antola Unit is the uppermost structural unit of the Northern Apennines orogenic stack of the east Liguria region (Figure 1), and it tectonically lies above the internal Ligurian Units (i.e., the Gottero and Vara Units Auct. [17]). This uppermost structural position can be compared to that of the "Helminthoid Flysch" of the Western Liguria Flysch (i.e., the San Remo Flysch) that thrusts westward onto the Provençal and Briançonnais Units (Figure 1) [18–26].

The Antola Unit, which crops out along the coast between Chiavari and Genova and also continues inland of the Ligurian Apennine, consists of three formations (in stratigraphic order) [3]: (i) the Montoggio Shale (Campanian, [17,27]), represented by multicolored shales with rare intercalations of quarzitic sandstones, arenaceous limestones, and marly limestones; (ii) the approximately 1000 m thick Mt. Antola Limestone (also known as the Mt. Antola Formation (late Cretaceous–?Paleocene [17,27,28]), mainly composed of turbiditic marly limestones, calcarenites, and calcareous-marls with local centimetric shale intercalations that were deposited in an abyssal plain environment below the carbonate compensation depth (CCD) [29]; and (iii) the Pagliaro Claystone (Late Cretaceous–late Paleocene [30]), mainly composed of thick shale beds alternating with siliciclastic, thin bedded turbidites and minor calcareous turbidites deposited in a deep marine environment [30].

A few structural studies were carried out on the Antola Unit [3,31].

The author of [31] identified for the Antola Unit four deformational events: the first one occurred between the early Eocene and early Oligocene, while the others were ascribed to the Miocene–Pliocene time interval. The Antola Unit did not suffer metamorphic event as it is pointed out by the study of illite crystallinity that remains in the diagenetic field [32].

In particular, between Chiavari and Genova, [3] recognized two main tectonic events. The first one was a polyphase brittle–ductile event, whereas the second was a brittle event that also can be identified in part in the PC. The first event was characterized by five tectonic phases (F1 to F5). At least the first three tectonic episodes took place before the deposition of the Portofino Conglomerate. The first phase (F1) was characterized by rare folds with NW/SW vergence; the second (F2) was related to folds with SE vergence; F3 was associated with subvertical conjugate shear bands; F4 was characterized by low-angle shear planes with a top-to-the-NE/NW component of shear; and F5 was related to folds with their subvertical non-coaxial planar cleavage (transected cleavage folds [33]). The first three tectonic phases identified in the Antola Unit should be ascribed to the pre-Oligocene Alpine phase in agreement with what was defined by [11,14,34] in the surrounding areas.

Instead, a greater number of studies were carried out regarding the structural features of the underlying Internal Ligurids (i.e., the Gottero, Vara, Bracco/Val Graveglia, and Colli/Tavarone Units [11–14,34,35]) that revealed a different structural history characterized by at least three deformational events of late Paleocene–middle Eocene age.

The PC stratigraphically rests on a polydeformed Helminthoid Flysch Unit (Figures 2 and 3a,b) commonly referred to the tectonostratigraphic Antola Unit [17] through an erosional angular unconformity. The authors of [34] interpreted its substrate as belong-

ing to a distinct Helminthoid Flysch tectonic Unit (UA3) that thrusted eastward onto the tectonic pile of the Antola Unit.

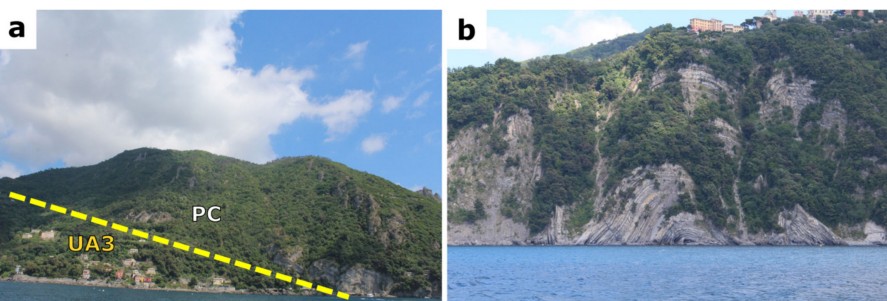

**Figure 3.** (**a**) Angular unconformity (dashed yellow line) between the highly folded UA3 Helminthoid Flysch and the overlying east-dipping Portofino Conglomerate (PC) at Porto Pidocchio (seen from the SW); (**b**) polyphase folding in UA3 (seen from the SW).

The PC, the maximum thickness of which is about 600 m (see geological cross section in Figure 2), mainly consists of massive or poorly stratified conglomerates with few intercalated sandstone and siltstone beds that are more frequent in the upper part of the succession [1–4]. As a whole, these conglomerates are moderately to poorly sorted and clast- to matrix-supported with a sandy to sandy-silty matrix and a carbonate cement; the clasts show rounded/subrounded to subangular shapes and variable dimensions (generally from a few centimeters to a few decimeters and exceptionally up to one meter) [4].

Their polygenic skeletal fraction, consisting mainly in sedimentary rocks and high-pressure–low-pressure (HP-LP) and medium-temperature–high-temperature (MT-HT) metamorphic rocks, was first defined in [1,2,4]. The authors hypothesized a conglomerate source from the Alpine Corsica. The same authors considered the PC as an Alpine post-orogenic marine basin located between the Ligurian Alps and the Apennine realm.

Locally, the Quaternary covers (consisting in alluvial deposits, debris, and landslides) are present especially in the areas of the contact between the Helminthoid Flysch and the PC.

## 4. Lithostratigraphy, Sedimentary Features, and Petrofacies of the PC

The unconformable contact between the PC and the Ligurian substrate (Helminthoid Flysch, UA3) is rarely exposed because of the dense vegetation cover, and it is observable only in a few places (e.g., Porto Pidocchio; Figure 3a). Locally, fractures and high-angle normal faults that displaced UA3 are sealed and partially filled by the deposition of the PC conglomerates (Figure 4). In addition, a post-depositional weak wide-range folding and the following neotectonic high-angle normal faulting [36] deformed the PC and its substrate (see geological cross section in Figure 2).

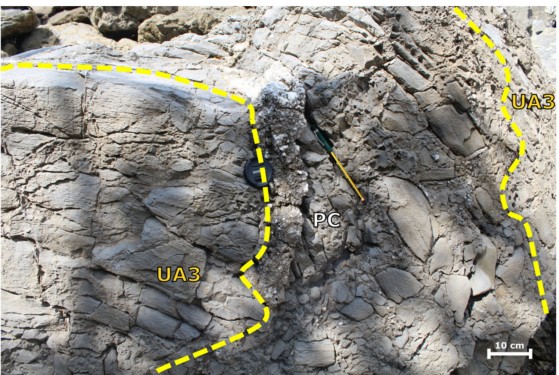

**Figure 4.** PC conglomerates fill a fracture in the UA3 underlying succession at Porto Pidocchio (seen from the SW).

Lithological–sedimentological aspects of the conglomerates and the compositional nature of their clasts allowed us to distinguish three different litho/petrofacies (see also [4]); from bottom to top (Figure 5): (i) Paraggi litho/petrofacies (fP); (ii) Monte Pallone litho/petrofacies (fMP); and (iii) Monte Bocche litho/petrofacies (fMB). Each facies grades vertically into the following one.

**Table 1.** Percentage composition of clasts in the three litho/petrofacies of the PC. Nature of the clasts: CA: carbonate sedimentary clast; MB: metabasite clast; MC: metachert; MCS: metamorphic carbonatic clast; QS: metasiliciclastic clast; PTV: volcanite clast; HM: high-medium-temperature metamorphic clast.

| Litho/Petrofacies | Thickness (m) | CA% | MB% | MC% | MCS% | QS% | PTV% | HM% | Fossils in Clasts | Fossils in Matrix |
|---|---|---|---|---|---|---|---|---|---|---|
| Bocche | 150 | 10 | 0 | 0 | 5 | 4 | 1 | 80 | No | No |
| Pallone | 60 | 30 | 20 | 5 | 30 | 15 | 0 | 0 | No | No |
| Paraggi | 250 | 96 | 4 | 0 | 0 | 0 | 0 | 0 | Globotruncanids Heterohelicids Calisphaeres Discocyclinids Nummulitids Crinoid fragments Sponge spicules Red calcareous algae | Globotruncanids Heterohelicids Nummulitids Orthophragmines Red calcareous algae Echinoids plates and spines Bryozoans Mollusk fragments Worm tubes Sponge spicule Textulariids Rotaliids Nodosariids Globigerinids |

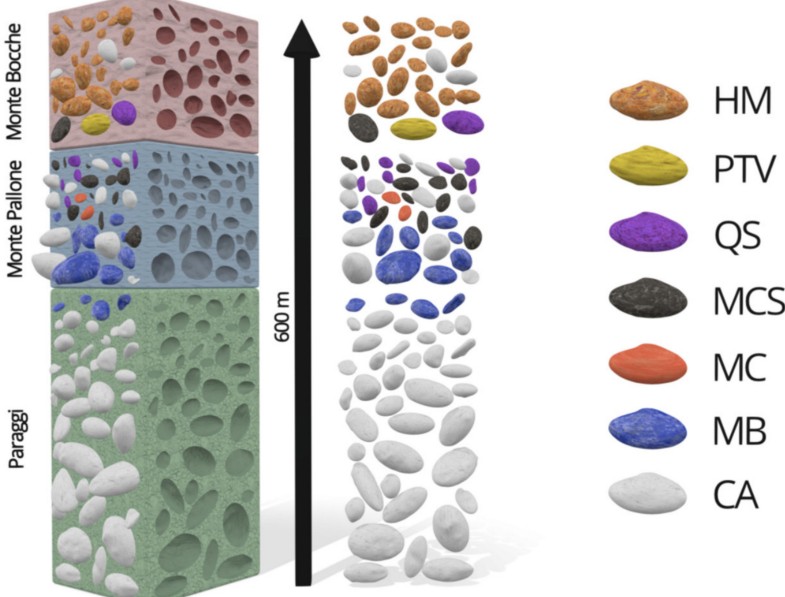

**Figure 5.** Stratigraphic succession of the litho/petrofacies in the PC. Nature of the clasts: CA: carbonate sedimentary clast; MB: metabasite clast; MC: metachert clast; MCS: metamorphic carbonatic clast; QS: metasiliciclastic clast; PTV: acidic volcanite clast; HM: high-medium-temperature metamorphic clast. The distribution of the different clasts in the figure reflects the percentages reported in Table 1.

The author of [4] described in detail the nature of the clasts in the three different lithofacies. In this paper, the original data were reprocessed with a focus on the most significant typology of the clasts in terms of provenance.

Here, we considered the following lithologic–petrographic groups of clasts: (a) CAs: carbonatic sedimentary rocks, including calcilutites, calcisiltites, calcarenites, calcirudites, marls, marly limestones, and biosparites/biomicrites characterized by the occurrence of globotruncanids, heterohelicids, calcisphaeres, crinoid fragments, and sponge spicules;

(b) MBs: high-pressure–low-pressure metamorphic basic igneous rocks characterized by coarse metagabbros, metaophiocalcites, blue meta-pillow lavas, porphyritic diabases, and rare meta-hyaloclastites; (c) MCs: metacherts, including metamorphic radiolarian cherts and radiolarites in which radiolarians are deformed but preserved; (d) MCSs: meta-sedimentary carbonates, including calcschists, marbles, and dolomitic marbles; (e) QSs: quartz-dominated (meta)siliciclastic rocks, including quartz clasts, quartzites, green or graphitic phyllites, anagenites, green/whitish quartz arenites, and quartz conglomerates; (f) PTVs: acidic volcanic rocks; (g) HMs: regional-type metamorphic rocks in amphibolitic conditions, including micaschists, paragneiss, orthogneiss, amphibolites, and migmatitic gneiss.

For percentage and schematic vertical distribution of the clast associations in the different litho/petrofacies, see Table 1 and Figure 5.

### 4.1. The Paraggi Litho/Petrofacies (fP)

This litho/petrofacies represents about 250 m of the succession that is characterized by dominant, mainly matrix-supported, and mostly inversely graded conglomerates with a very poorly sorted and coarse-grained matrix. Clasts are generally rounded/sub-rounded and characterized by an overall bladed-discoidal shape and a sphericity of 0.4–0.6 (Figure 6a–g).

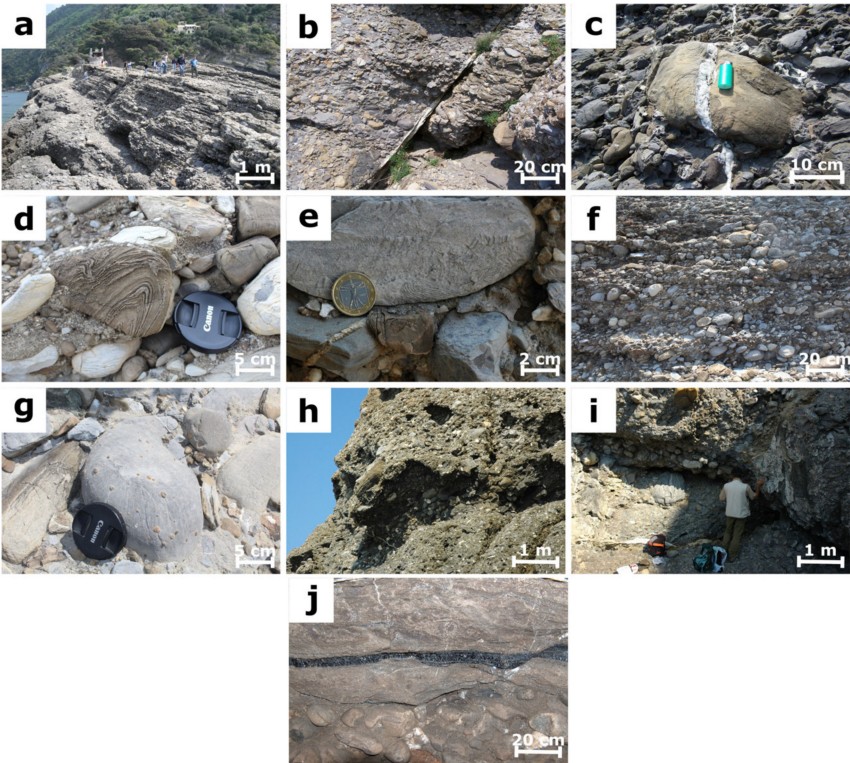

**Figure 6.** (**a**) Outcrops at Punta Chiappa (seen from the NW); (**b**) fracture in the Paraggi litho/petrofacies (seen from the SW); (**c**) details of clasts crossed by fractures with white calcite veins (seen from the E); (**d**) clast of marly limestone with convolute laminations (seen from the NW); (**e**) clast of limestone with *Nereites irregularis* (seen from the NW); (**f**) SW-dipping imbricated CA clasts (seen from the SW); (**g**) limestone clasts with crinoids (seen from the NW); (**h**) reverse-graded bedding (seen from the SW); (**i**) normal-graded bedding (seen from the SW); (**j**) coal-rich lens inside an arenaceous bed (seen from the N).

Subordinately, there may be locally present bodies of: (i) clast-supported conglomerate beds with well-rounded/rounded, often discoidal clasts and rare weakly laminated sandstone and pebbly sandstones as centimetric to decametric intercalations; and (ii) matrix-supported breccias and immature conglomerates (with angular to subrounded

clasts) generally showing a chaotic texture or inverse (rarely normal) grading that are prevalent in the basal part of this lithofacies (Figure 6h,i).

The geometry of the beds is roughly planar at the scale of the outcrops (Figure 6a); erosive structures are common at the base of the conglomeratic beds (Figure 6h,i) with local scours that are decimetric in depth and sometimes filled with clast-supported coarse conglomerates. Subangular to subrounded pebbles/cobbles to rare boulders of fragmented Helminthoid Flysch beds (with a max thickness of 3–4 m) unevenly occur at places in the chaotic basal part (Figure 6e; see also Figure 5 in [1]). Locally, the conglomerate clasts are imbricated or lie parallel to the bedding (Figure 6f). Moreover sandy "rip-up" clasts can be locally found within the conglomeratic beds.

This conglomeratic lithofacies appear locally bedded, particularly in the Punta Chiappa area (Figure 6a). The bed thickness ranges from 50 cm to a few tens of meters, but the thicker strata appear often as the result of amalgamation processes of different beds (Figure 6a). The presence of fractures and calcite veins that cut and locally displaced the PC (Figure 6b,c) is noteworthy. Centimetric (max 8 cm thick) coal-rich levels are locally found embedded in arenaceous intercalations (Figure 6j).

The paleocurrent data obtained from the 95 poles of the imbrication surface of the clasts indicate a flow in a NW–SE direction with a likely NW source with respect to the present geographic references (Figures 6f and 7).

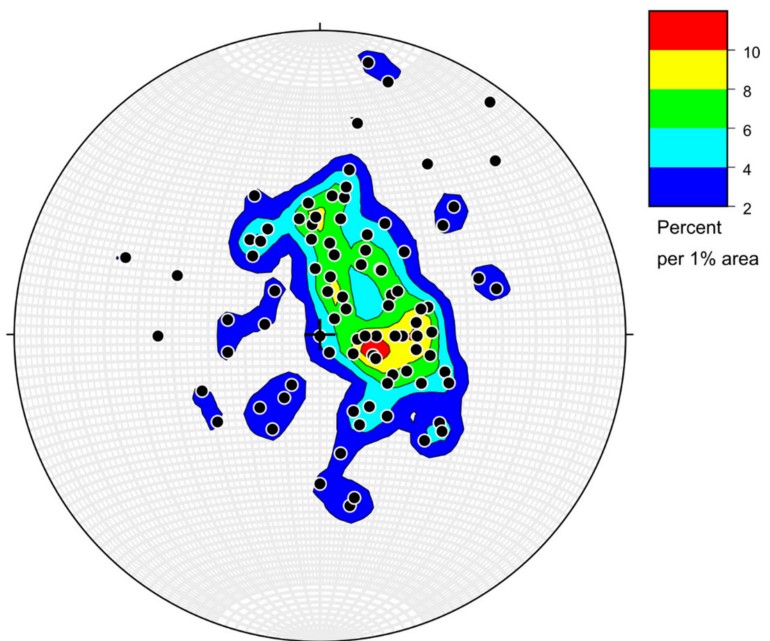

**Figure 7.** Stereonet of the poles of the imbrication plane of the clasts in the Paraggi litho/petrofacies (equal area lower hemisphere; 95 poles).

Most parts of this litho/petrofacies are characterized by the large dominance of carbonatic clasts (CAs; Figure 6d,e,g); rare metabasic igneous rocks (MBs) associated with CAs are present only in the top part. The authors of [1] also found biocalcirudite clasts with discocyclinids, nummulitids, and red calcareous algae.

The matrix is represented by a calcareous sandy-silt including a very low percentage of siliciclastic grains (i.e., quartz, albite, serpentine, clinochlore, and muscovite); few grains of dolomite are also present. A diffractometric analysis [4] pointed out that it is mainly composed of 65% calcite, 28% quartz, 4% albite, and 3% clinochrysotile.

The carbonate grains of the very coarse sandy matrix are biosparites and biomicrites in which Late Cretaceous fossils (i.e., globotruncanids, heterohelicids, etc.) are commonly associated with abundant sponge spicules and crinoid fragments (Figure 8a). Other calcareous components are fragments of middle Eocene fossils such as nummulitids

and orthophragmines (Figure 8f) and locally abundant fragments of coralline algae thalli (Figure 8h). Well-preserved marine macro- and microfossils were also found in the matrix of the conglomerates for the first time. The most common taxa are echinoid plates and spines, bryozoans, mollusk fragments, worm tubes, sponge spicule, benthic and planktonic foraminifera (i.e., textulariids, rotaliids, nodosariids and globigerinids) and coalified materials (Figure 8). In particular, three specimens of planktonic foraminifera are present within the matrix. The specimen in Figure 8b (exposed through its axial section) is characterized by a width of 490 μm and a height of 380 μm. The test is spinose with cylindrical pores, possibly cancellate. For what it is visible on the section, it appears slightly elongate but subrectangular in outline. The chambers are rounded but compressed, and the visible suture is depressed and vertically dividing the last chambers. The early chambers are partially visible, but three whorls can be recognized (with the first two coiled in a low trochospire), whilst the last whorl is extremely high and fast embracing the previous chambers. These characteristics point to *Globigerinatheka index,* which indicates an upper Lutetian to uppermost Priabonian age. The specimen in Figure 8c has the same structural element of the previous one but is slightly smaller in dimension. Its diameter is only 350 μm but still within the population limits of *G. index* as reported in [37]. Other possible *G. index* are reported in Figure 8d,e.

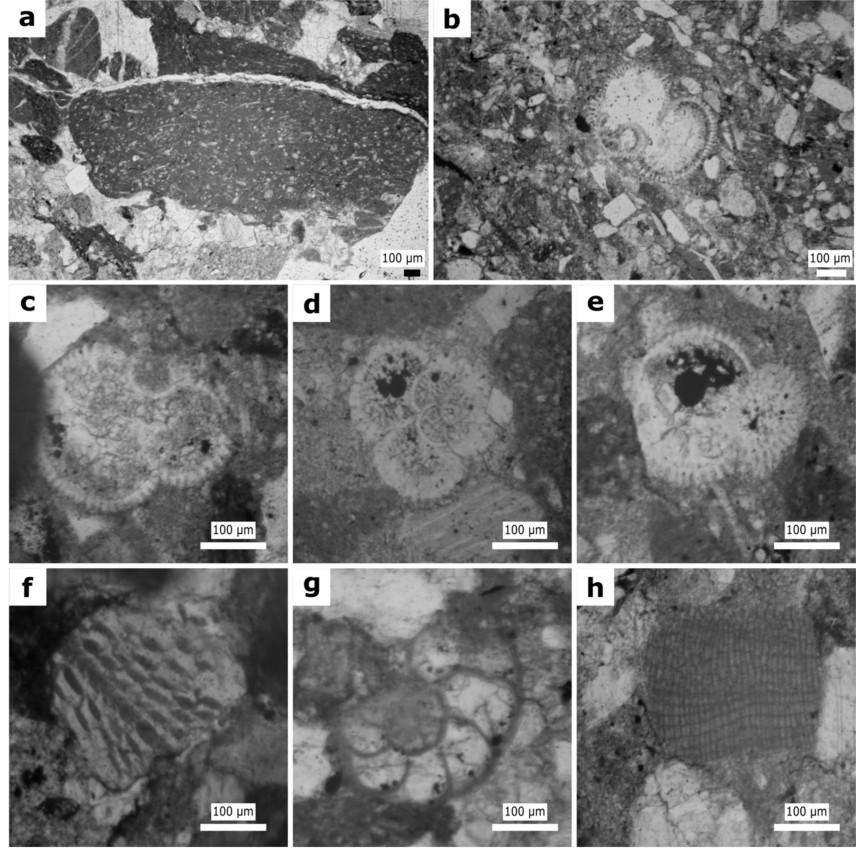

**Figure 8.** (**a**) Clast of Helminthoid Flysch with crinoid fragments and sponge spicules; (**b–e**) *Globigerinatheka index*; (**f**) Orthophragmine; (**g**) Rotaliid foraminifer; (**h**) Coralline algae thallus fragment.

### 4.2. The Monte Pallone Litho/Petrofacies (fMP)

This litho/petrofacies represents approximately 60 m of the succession. It is characterized by poorly stratified conglomerates and minor sandstone intercalations (Figure 9a). The bed thickness, when measurable, ranges from 60 cm to 7 m; amalgamated beds commonly occur. The conglomerates are poorly sorted (granule to cobble) with angular/subangular to subrounded clasts and a sandy-silty matrix and carbonate cement. The shape and

sphericity of the clasts depends on their lithology: equant-bladed to discoidal-elongated and 0.5–0.6, respectively, for silicate rocks; generally discoidal/bladed and 0.5–0.6 for the crystalline carbonates; and from equant-bladed to discoidal and 0.6–0.7 for the metabasic rocks. The ruditic beds, which are generally massive (often with an internal chaotic texture) or sometimes characterized by a crude normal or reverse grading, are separated by erosional contacts. Pebble clusters can be locally identified in the massive beds that sometimes are particularly rich in a sandy matrix (Figure 9b). Some clast-supported gravel beds with rounded/well-rounded clasts characterized by basal erosive structures are recognizable in this lithofacies as well. Sandy lenses rarely occur below the erosive beds or at the wavy top of the clast-supported layers. No fossils were found.

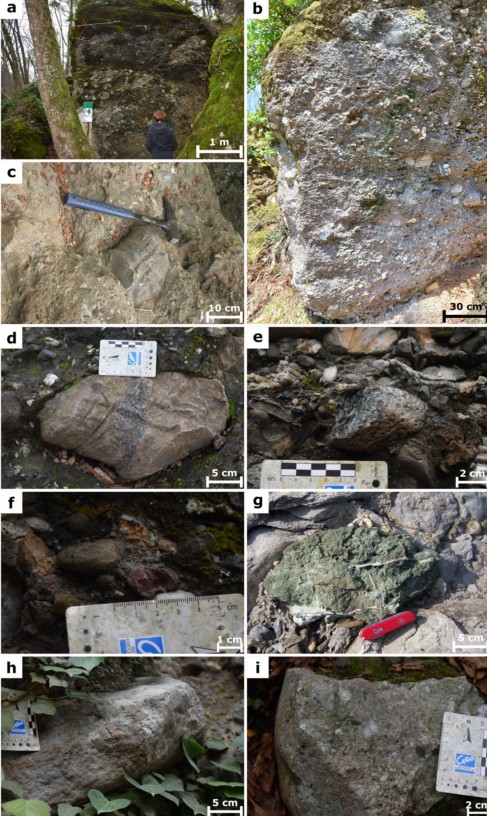

**Figure 9.** (**a**) Outcrop of the Monte Pallone litho/petrofacies (Loc. Pietre Strette; seen from the S); (**b**) pebble clusters in a matrix-rich, chaotic, and massive bed (seen from the W); (**c,d**) marble clast (MCS; (**c**): seen from the NW; (**d**): seen from the SW); (**e**) metagabbro clast (MB; seen from the W); (**f**) metachert clast (MC; seen from the E); (**g**) pillow lava (MB; seen from the S); (**h,i**) quartzite clasts (QS; (**h**) seen from the N; (**i**): seen from the N).

The 90 poles of the imbrication surface of the clasts suggest a NE–SW directed flow with a likely SW source with respect to the present geographic references (Figure 10).

The framework of the Monte Pallone litho/petrofacies conglomerates is characterized by the following types of clasts: marbles, calcschists, and metacherts (MCS +/− MCs; Figures 9c,d,f and 11a,d); carbonates (CAs); high-pressure effusive and intrusive basic rocks (MBs; Figure 9e,g); and quartz, quartzites, green or graphitic phyllites, anagenites, green/whitish quartz arenites, and quartz conglomerates (QSs; Figures 9h,i and 11b). The interstitial fraction is composed of a quartz- and serpentine-rich sandy-silty matrix and carbonate cement. A low percentage of other elements (i.e., carbonate grains, albite, clinochlore, dolomite, and clay minerals) are also present. No fossils were found. Diffractometric analysis of the litho/petrofacies matrix in the Monte Pallone [4] pointed out that it is composed of 32% calcite, 28% quartz, 25% chrysotile, 8% clinochlore, 3% dolomite, 2% albite, and 2% nontronite.

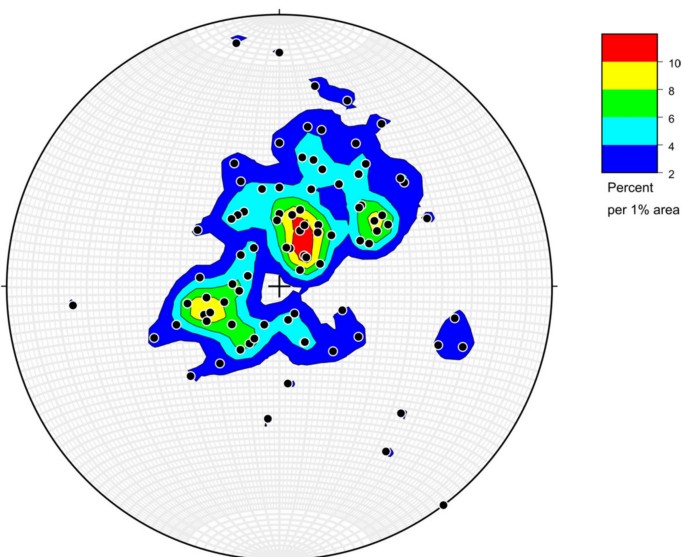

**Figure 10.** Stereonet of the poles of the imbrication plane of the clasts in the Monte Pallone litho/petrofacies (equal area lower hemisphere; 90 poles).

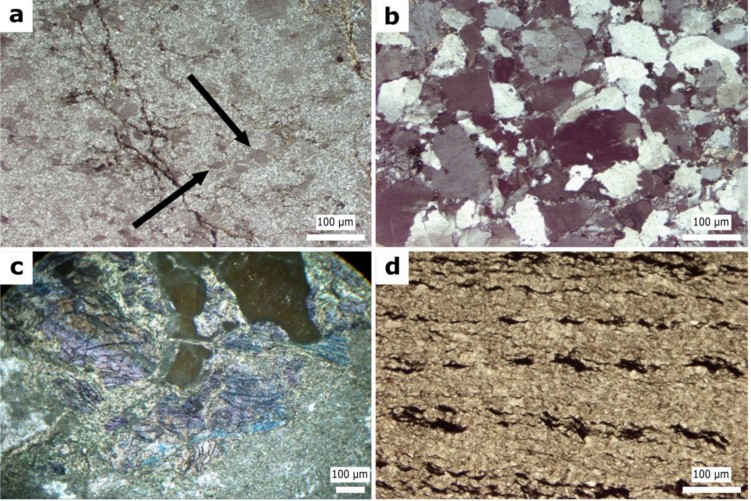

**Figure 11.** Photomicrographs of the metamorphic clasts within the Monte Pallone litho/petrofacies: (**a**) MC metachert with ghosts of deformed radiolarians identified by black arrows (parallel polars); (**b**) QS metasiliciclastic clast (crossed polars); (**c**) blue amphiboles in MB metagabbro (crossed polars); (**d**) MCS calcschist (crossed polars).

The petrographic studies of the basic rocks revealed high-pressure mineralogical assemblages formed by sodic blue amphibole + pumpellyite +/− brown amphibole +/− tremolite-actinolite + oxides +sphene [4] (Figure 11c).

Metacherts show a penetrative secondary anisotropy that deforms the Radiolaria as ellipsoids with the major axis parallel to the secondary anisotropy (Figure 11a). Locally, the metacherts show ultramilonitic levels characterized by a very fine grain size.

### 4.3. The MonteBocche Litho/Petrofacies (fMB)

This lithofacies is unfossiliferous and is represented by an up to 150 m thick succession of alternating tabular bedded, matrix-supported, moderately sorted fine and medium-grained conglomerates (Figure 12a,b) grading upward into coarse-grained clast-supported conglomerates. Similarly to Monte Pallone, the clasts are composed of regional metamorphic rocks. They exhibit an equant-bladed to discoidal shape and a sphericity of 0.5–0.7.

The thickness of the beds ranges from 50 cm to 2.5 m (Figure 12a,b). The internal texture of these rudites is massive and chaotic and includes boulders in the middle and upper part of the beds, but inversely graded beds are also present. The beds of this lithofacies are characterized by an overall reduction in the dimensions of the clast and in their quantity in respect to the matrix so that pebbly sandstone beds are relatively frequent at places. This aspect highlights a possible gross overall "fining-upward" trend of the Portofino Conglomerate. In some cases, the pebbles are aligned parallel to the bedding and are often lenticular. Clast-supported levels may be present at the top of the matrix-supported beds (Figure 12b). Moderately to well-sorted, coarse-grained sandstone lenses may be locally interbedded as well.

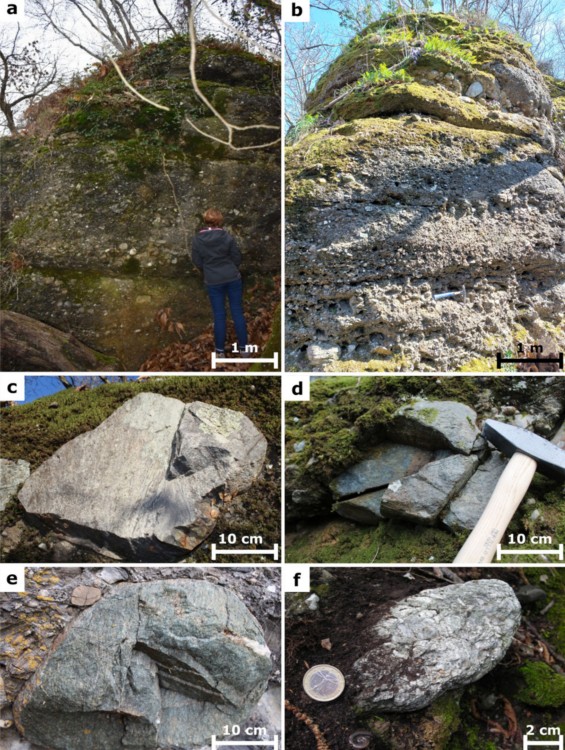

**Figure 12.** (**a**) Outcrop of the Monte Bocche litho/petrofacies in the Monte Bocche locality (seen from the E); (**b**) matrix-supported, fine- and medium-grained conglomerate beds with a lenticular clast-supported, coarse-grained conglomerate level in the upper part (seen from the N); (**c**) clast of migmatitic HM gneiss (seen from the N); (**d**) HM clast of amphibolite (seen from the N); (**e**) HM clast of micaschist (seen from the N); (**f**) HM clast of metagranitoid (seen from the NW).

The 150 poles of the imbrication surface of the clasts point to a NE–SW direction of the flow with a SW source with respect to the present geographic references (Figure 13).

The framework fraction is mainly composed of medium-temperature–high-temperature metamorphic rocks (HM; Figure 12c–f), CA, and MCS with minor PTV and QS clasts. The peculiar occurrence of HM clasts that are absent in the underlying litho/petrofacies is noteworthy.

The interstitial fraction is composed of a quartz-dominated sandy-silty matrix with a low percentage of carbonate grains, albite, clinochlore, muscovite, serpentine, and clay minerals and minor carbonate cement. A diffractometric analysis [4] defined a composition of 70% quartz, 8% Fe-clinochlore, 8% nontronite, 7% clinochrysotile, 4% albite, and 3% muscovite.

The microscopic analyses performed on the most common HM clasts showed that they are composed of the following rocks: kyanite + garnet-bearing micaschists, mylonitic micaschists (Figure 14b), paragneiss, mylonitic paragneiss, stromatitic gneiss (Figure 14a), amphibolites, and two muscovite-bearing orthogneiss (types I and II [38,39]). The micaschists/paragneiss show a mineralogical assemblage formed by quartz + plagioclase +

muscovite + biotite + chlorite + apatite +/− garnet +/− kyanite. The mylonitic structure is frequent and is characterized by muscovite mica fish [40] and quartz microstructures such as deformed lamellae (recognizable at T: 550–600 °C and P: 3 kbar [41]), grain boundary asymmetric bulges (related to both a sliding and a non-uniform intergranular strain [42]), assemblage of elongated grains of constant size (type B2 [40]), and elongated grains parallel to the ribbon (type B3; recognizable in an amphibolitic condition). Stromatitic gneiss show a mineralogical assemblage composed of K-feldspar + quartz + biotite + muscovite + fibrolitic sillimanite + plagioclase +/− garnet. The two-micas bearing orthogneiss are characterized by a K-feldspar + quartz + biotite + phengite + plagioclase + epidote + apatite + chlorite mineralogical assemblage (Figure 14c). The rare amphibolites are characterized by a hornblende + sphene + plagioclase assemblage. (Figure 14d).

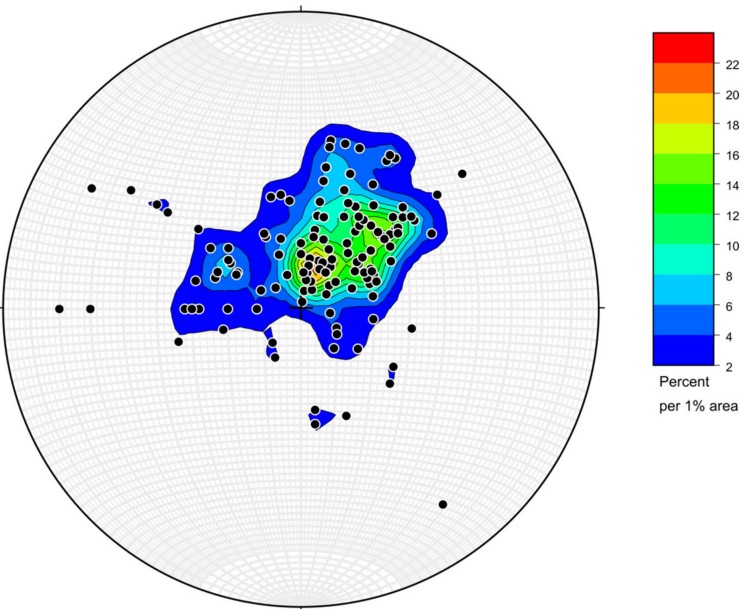

**Figure 13.** Stereonet of the poles of the imbrication plane of the clasts in the Monte Bocche litho/petrofacies (equal area lower hemisphere; 150 poles).

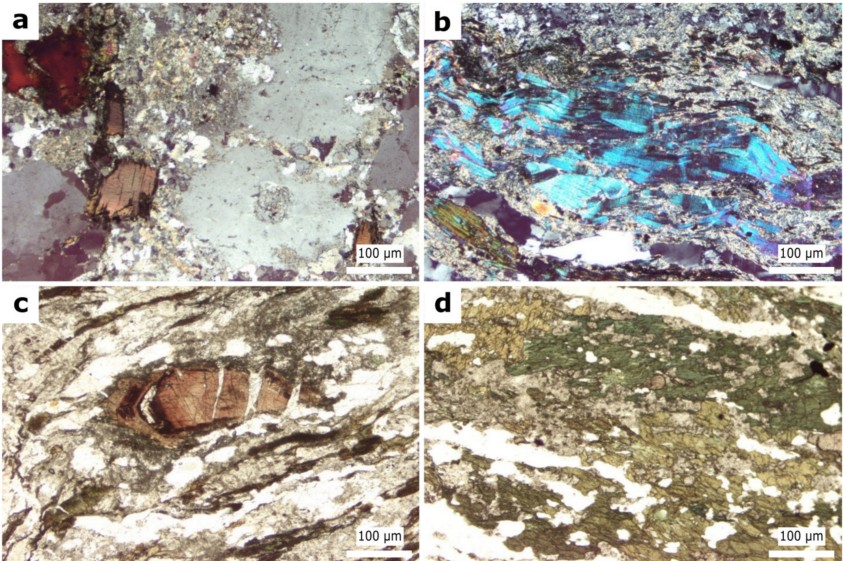

**Figure 14.** Photomicrographs of the metamorphic clasts within the Monte Bocche litho/petrofacies. (**a**) HM migmatitic gneiss (crossed polars); (**b**) HM mylonitic micaschist (crossed polars.); (**c**) zoned allanite in metagranitoids (parallel polars); (**d**) HM brown/green hornblende in amphibolite (parallel polars).

The SEM analyses of the white micas [4] allowed us to distinguish two types of phengite on the basis of Si vs. $Al_{tot}$ [43]: (i) 6.251–6.995 atoms per formula unit (p.f.u.) of Si, 1.005–1.749 p.f.u. of Al(IV) (value of Si vs. $Al_{tot}$ comparable to the pre-Alpine phengite described by [43]), 1.25–9.32 wt% of FeO, 1.76–4.03 wt% of MgO, 0.19–2.75 wt% of $TiO_2$; (ii) 5.969–6.077 p.f.u of Si., 1.923–2.031 p.f.u. of Al(IV) (value of Si vs. $Al_{tot}$ comparable to the Alpine phengites described by [43]) and 1.19–1.43 wt% of FeO, 0.96–1.76 wt% of MgO, 0.06–1.32 wt% of $TiO_2$.

The epidote was characterized by 0.33 wt% of MgO, 0.33 wt% of MnO, and 1.62 wt% of $TiO_2$.

## 5. Discussion

For a better comprehension of the cited geological units and formations included in this chapter, the authors refer to [6].

The sedimentological features of the three lithofacies allowed their attribution to deposition from hyperconcentrated flows to gravelly high-density turbidity currents (cohesive flow and grain flow deposits in Mutti's Facies F2 and F3 [44]) in an apical part of a submarine fan. These deposits testify to an abundant sedimentary input from a relatively close source area in strong erosion conditions, probably due to a strong tectonic activity.

The vertical variation in the sedimentary inputs may probably be related to the progressive dismantling of source(s)/area(s) that experienced a coeval tectonic uplift, allowing the erosion of its different nappes.

Previous hypotheses regarding the feeding area of Portofino Conglomerate identified the source in the Alpine Corsica [2] or interpreted it as an Oligocene fan-delta in front of a mountain range located south of the present-day Genova–Savona coastline [45].

Our data on the composition of the conglomerate frameworks allowed a more precise localization of the source area(s) of the PC conglomerate while also taking into consideration the middle-late Eocene paleogeography of the circum-Mediterranean area. The data allowed us to restrict the source area west of the Voltri Unit (belonging to the Metaophiolite Units in Figure 1) because of the absence of HP-LT serpentinites and metaultrabasic-rock-derived clasts that are typical of that area. In addition:

(a) The CA biocalcirudite clasts containing discocyclinids, nummulitids, globorotaliids, and red calcareous algae found in [1] in the Paraggi litho/petrofacies might have been supplied by the "Nummulitique" of the Provençal Domain or by the lithostratigraphic top of the Briançonnais Units [18,20–26,46,47]; the other dominant CA carbonate clasts were likely supplied by an Helminthoid Flysch sedimentary succession characterized by the presence of Upper Cretaceous biomicrite/biosparite limestones with globotruncanids, heterohelicids, calcisphaeres, and sponge spicules. These lithotypes appear to be comparable to those described for the Western Liguria Flysch complex (i.e., the San Remo Flysch) [22,46] but not with those of the Antola Unit in the Genova–Chiavari area resting below the UA3 tectonic unit [48], where these microfacies are uncommon.

(b) The MB clasts may be related to the lithotypes described for the Metaophiolite Units (i.e., the Montenotte Unit) [28,49,50] as well for the metagabbro/metaophicalcites/prasinites clasts [49,51,52].

(c) The MC clasts may be related to the metacherts of the Metaophiolite Units (i.e., the Montenotte Unit) belonging to the oceanic Piedmontese domain, which exhibits comparably deformed but preserved radiolarian faunas [32,49,51,52].

(d) The MCS marble clasts are similar to the "Calcaires Pointillés" [53] occurring in the Metaophiolite Units (i.e., the Montenotte Unit) [32,49,51,52] or to the metadolostones and dolomitic marbles recovered in the Piedmontese Units (i.e., Angassino-Terma, Gazzo-Isoverde, and "Trias-Lias" Monte Sotta Units) [28,49,50]. Dolomitic rocks can be correlated with similar lithotypes of the Briançonnais Units (i.e., the "Trias-Lias unit") described in [32,52].

(e) The QS clasts are texturally and compositionally very similar to the lithotypes of the Permian "Verrucano" and lower Triassic siliciclastic deposits of the Ligurian Briançonnais Units [54].

(f) The scarce PTV clasts should derive from the internal Ligurian Briançonnais Units (i.e., the Pamparato–Murialdo Unit) that are characterized by a reduced volcanic and carbonate succession [18,54].

(g) The petrographic and structural features of the HM clasts [4] could be relatable with the micaschist/paragneiss successions outcropping in the Briançonnais Units (i.e., the Calizzano–Savona Unit) described in [38,39,43,55,56]. In addition, the SEM mineralogical compositions of the epidotes and phengites (showing a pre-Alpine amphibolite facies metamorphic imprint and a possible Alpine recrystallization as well as those described in [43,55]) in the polymetamorphic two micas bearing orthogneiss clasts [4] are similar to that obtained in [43,55] in the Briançonnais Units (i.e., the Calizzano–Savona Unit).

To summarize, the geology of the feeding area(s) can be reconstructed through the composition of the three litho/petrofacies characterized by: (i) CA, (ii) MB + MCS + MC + QS, and (iii) HM + PTV clasts.

The first group (CA) may be supplied by a tectonostratigraphic unit similar to the Western Liguria Flysch (i.e., San Remo Flysch Unit), the second one (MB + MCS + MC + QS) appears to be derived from a tectonometamorphic unit comparable to the Metaophiolite Units (i.e., the Montenotte Unit), while the third group (HM + PTV) may be the result of the erosion of more internal Ligurian Briançonnais Units (i.e., the Pamparato-Murialdo and Calizzano-Savona Units). So, the vertical variation in the nature of the clasts in the PC suggests a progressive erosion of a source area that can be identified in a typical nappe stack in the Ligurian Alps.

The PC stratigraphically rests onto a polydeformed Helminthoid Flysch Unit (UA3) [48] affected by a previous brittle extensional tectonic activity (see the cross section in Figure 2) and characterized (similarly to the San Remo Flysch and Western Liguria Flysch) by the occurrence of biosparite/biomicrite lithotypes that constitute the CA clasts in the basal Paraggi litho-petrofacies of the PC. It is important to remember that these lithotypes seem to be rare in the Antola Unit on which the UA3 unit thrusted with the overlying PC sedimentary cover.

Therefore, the provenance area of the PC and its UA3 substrate had to be located in the Western Liguria—west of Voltri Unit (see [49]) where tectonic units similar to those recorded in the clasts of the PC form part of the Ligurian Alps (for a geologic view of the considered area, see [6,19–26,28,46,49]).

As regards the age of the PC, we should consider that the HP metamorphism of the Ligurian-Piedmontese Oceanic Units in the Ligurian Alps has an Eocene age in the Voltri Massif (45–39 Ma in [57]; 49.68–38.15 Ma in [58]; 50–38 Ma in [59]; 40.6–37.7 Ma in [60]) and in Corsica (45–37 Ma in [61,62]). This constitutes a lower time limit for the production of the metamorphic clasts of the PC from the already recrystallized and exhumed HP units. Nevertheless, the age is still debated because of the scarcity of fossils; [1,2] proposed an early Oligocene age on the basis of the occurrence of bioclasts (i.e., bivalve fragments and globigerinids) in the matrix and of regional considerations (e.g., correlations with the epiligurian Ranzano sandstones and/or with the Tertiary Piedmont Basin [9,63]).

The well-preserved planktonic microfossils (i.e., *Globigerinatheka index*) found in the matrix of our samples from the Paraggi outcrop indicate a middle-late Eocene age.

*Globigerinatheka index* has a relatively broad stratigraphic distribution spanning from the E9 to E15 biozones [37,64–67] that indicates a time interval between 41.9 and 34.5 Ma (i.e., upper Lutetian to uppermost Priabonian). This interval is coeval with the first Cenozoic rotational phase of the Sardinia–Corsica–Briançonnais system (50–30 Ma [68–70]); this would explain why the paleocurrent data show an evident rotation of the flow of the detritic inputs from NW–SE (Paraggi litho/petrofacies) to NE–SW (M.Pallone and M.Bocche litho/petrofacies). Due to the lack of fossils in the uppermost two litho/petrofacies, we do

not rule out a possible younger age (early Oligocene?) for the end of the PC sedimentation that in any case must have occurred before the second Cenozoic rotational phase of the Sardinia–Corsica–Briançonnais system (21–16 Ma; [68–70]).

After the deposition of the Portofino Conglomerate on UA3, this assemblage likely moved from its original paleogeographic location to the east with a NE "Apennine" vergence and suffered a weak folding and faulting.

The significance of the PC basin in the geological evolution of the Ligurian Alps and of the eastern "Apenninic" Liguria can be highlighted in the following tectono-sedimentary hypothesis.

The tectonic–stratigraphic situation of the PC recalls in part that of the Epiligurian Units of the Northern Apennines that were sedimented within minor basins placed onto the Ligurian thrust sheets mostly during their eastward movement; this sedimentary contact corresponds to a regional angular unconformity of middle-late Eocene age [71–75]. Moreover, the Epiligurian depositional systems were likely laterally connected with the deposits of the episutural Tertiary Piedmont Basin [9,76]. In particular, the PC on UA3 resembles the deposition of the basal Epiligurian Unit in the Emilian Apennines (i.e., the Monte Piano Marls of late upper Eocene–early Oligocene age) onto the already deformed Helminthoid Flysch Units (i.e., the Antola Unit) and the External Ligurids (i.e., the Cassio Unit, Monghidoro Unit, and M. Morello Unit). The syn-orogenic Epiligurian basins are interpreted as episutural satellite basins (piggy-back or thrust top basins in [75,77–79]) and the northernmost prolongation of the Elba–Pianosa Ridge–Corsica basin system [80] (a thrust-top system on an accretionary wedge in [80]). In this frame, we interpreted the PC depositional system as a thrust-top basin that was active during the Eocene orogenetic events that occurred in the Ligurian Alps area (Figure 15). The PC episutural satellite basin was independent because clastic inputs from the coeval foredeep were created in the frontal part of the Ligurian Alps orogenic stack that was filled by the turbiditic deposits of the late Bartonian–early Priabonian Ventimiglia Flysch or Grés d'Annot (Provençal Unitsin Figure 1), the source area of which was located to the present south; i.e., the Corsica area [20–22,24,81–84].

After its sedimentation with at least a part of the basal UA3 substrate, the PC thrusted eastward onto the Antola Unit and then together onto the innermost part of Northern Apennines stack of nappes (e.g., the Gottero and Vara Units). Evidence of a thrust-top basin are described also in the Provençal area but attributed to the Pyrenean phase [85].

In this framework, is very difficult to identify the exact position of the PC basin in the Eocene–Miocene. The geodynamics of the Mediterranean area in this time interval was characterized not only by the last compressive Eocene–Oligocene phases and the beginning of the Miocene extension [86] but also by the counterclockwise rotation of the Sardinia–Corsica–Briançonnais system [68–70] and by the opening of the Ligurian–Balearic Sea in the upper Oligocene–lower Miocene [87]. This rotation was synchronous with and probably responsible for the documented N–S shortening that affected the Provençal area; it occurred simultaneously with the incorporation of the Briançonnais continental domain—likely connected to Corsica—into the western Alps [68].

Considering the existing bibliography on palaeomagnetic studies [68–70,88,89] between the Eocene and the present time, a ~95° counterclockwise rotation of the Sardinia–Corsica–Briançonnais system can be deduced. By adding this value to the data obtained by the paleocurrents, the source area of the PC at the Eocene–?Oligocene times was located in the northern areas where the Alpine Chain was under construction. In particular, the evident rotation of the original direction of the detritic inputs from NE–SW (Paraggi lithofacies) to NW–SE (Monte Pallone and Monte Bocche litho/petrofacies) can be related to a change in the location of the feeding area in different parts of the Alpine orogen (NE for Paraggi and NW for Monte Pallone and Monte Bocche) or (more probable) to the progressive counterclockwise rotation of the sedimentary basin due to the geodynamic evolution of this area.

In any case, the tectonosedimentary evolution of the PC can be likely interpreted as the innermost part of a submarine fan deposited in the northern continuation of the

wide basin that was the precursor of the Neogene Corsica Basin and fed by a mountain area developed along the Genova–Provençal area alignment and in connection with the meso-Alpine tectonic event or perhaps with the coeval Pyrenean Phase [68–70,81,90–92].

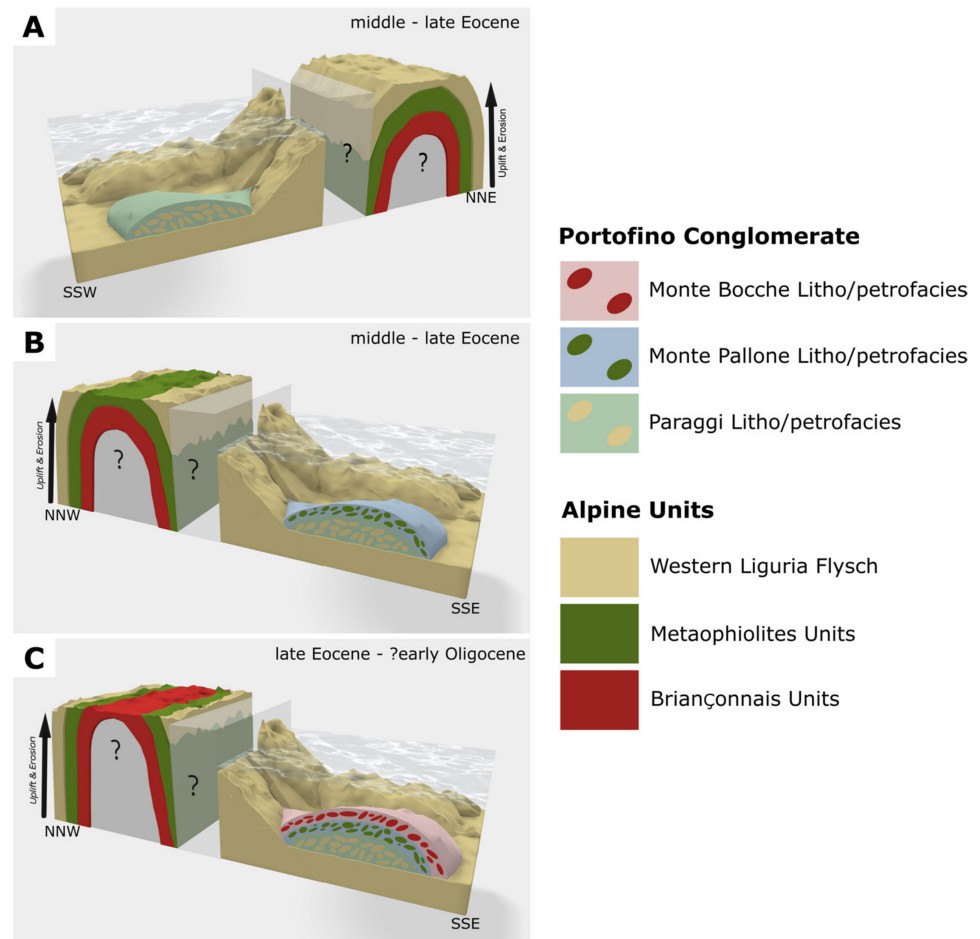

**Figure 15.** Evolutive hypothetical sketch of the PC sedimentary basin. (**A**) Beginning of erosion of the Western Liguria Flysch (i.e., San Remo Flysch) and the deposition of the Paraggi litho/petrofacies (middle-late Eocene); (**B**) erosion of the underlying Metaophiolite Units (i.e., the Montenotte Unit) with the deposition of the Monte Pallone litho/petrofacies after the first rotational phase of the Sardinia–Corsica–Briançonnais system (middle-late Eocene); (**C**) erosion of the Briançonnais Units (i.e., the Pamparato–Murialdo Unit and the Calizzano–Savona Massif) with the deposition of the Monte Bocche litho/petrofacies before the second rotational phase of the Sardinia–Corsica–Briançonnais system (late Eocene–?early Oligocene).

## 6. Conclusions

The approximately 600 m thick Portofino Conglomerate that sedimented on a polydeformed Helminthoid Flysch (UA3 unit) exhibits a clear vertical variation in sedimentary features and particularly in the composition of the skeletal fraction. This allows the distinction of three litho/petrofacies (Figure 5 and Table 1): Paraggi (lower part characterized by carbonate sedimentary clasts (CAs)), Monte Pallone (middle part characterized by metamorphic carbonatic clasts + metachert clasts + metabasite clasts + metasiliciclastic clasts (MCSs + MCs + MBs + QSs)), and Monte Bocche (upper part characterized by high-medium-temperature metamorphic clasts (HMs)). Clastic inputs may likely be supplied during the time by three distinct tectonostratigraphic/tectonometamorphic units piled up and cropping out in the Ligurian Alps (west Liguria): (i) the Western Liguria Flysch (i.e., San Remo Flysch); (ii) the Metaophiolite Units (i.e., the Montenotte Unit); and (iii) the Briançonnais Units (i.e., the Pamparato–Murialdo and Calizzano–Savona Units).

The vertical evolution of clast composition in the Portofino Conglomerate may be the result of a progressive deepening of the erosional level of the tectonic pile of nappes and/or of a syn-sedimentary tectonic segmentation of the provenance area in the Alpine belt under construction, which differentiated the feedings over time.

The Portofino Conglomerate sedimentary body can be considered as a depositional product of hyperconcentrated flows to gravelly high-density turbidity currents related to a coarse-grained apical part of a submarine fan that was active in the Ligurian Alps Domain (west of the Voltri Unit) during the middle-late Eocene to ?early Oligocene time interval as suggested by the well-preserved upper Lutetian to uppermost Priabonian planktonic foraminifers [93] occurring in the matrix of the basal Paraggi conglomerates.

Taking into account the nature of the clasts, the vertical evolution, the dynamics of sedimentation, and the paleogeographic reconstructions, the Portofino Conglomerate basin could have been located between Corsica (to the SW) and the Briançonnais/Western Ligurian Alps (to the N) (Figure 16).

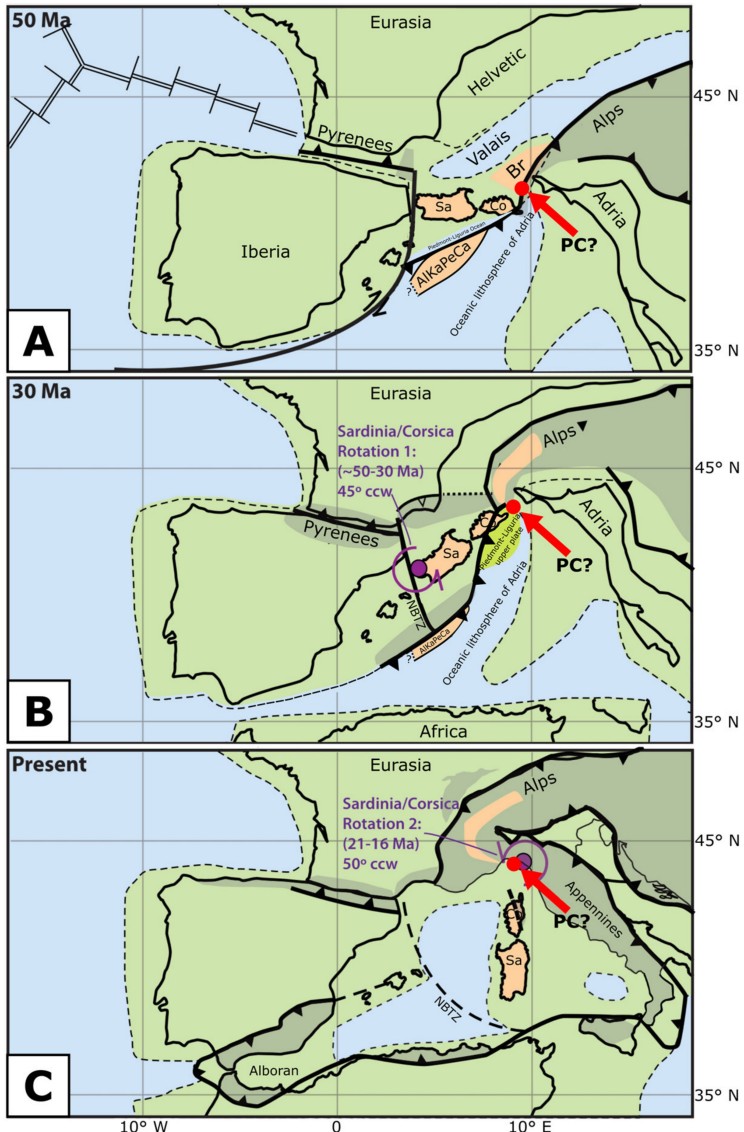

**Figure 16.** Hypothetical sketch of the central-western Mediterranean showing the position of the PC related to the rotation of the Sardinia–Corsica–Briançonnais systema at 50 Ma (**A**), at 30 Ma (**B**), and at the present time (**C**) (modified after [68,89]). Br: Briançonnais; Co: Corsica; NBTZ: North Balearic Transform Zone; PC: Portofino Conglomerate basin; Sa: Sardinia.

In this frame, the Portofino Conglomerate can more likely be attributed to the filling of a thrust-top basin that suffered—together with its substratum (UA3 unit)—a tectonic transport of at least 70 km from the Western Ligurian Alps to the present location in the innermost part of the Apenninic realm (likely during the Oligocene time). Another possible hypothesis that is much less probable is that the PC was deposited in a more distant pull-apart basin connected to the Pyrenees tectonic activity (i.e., the Pyrenees Phase) that later suffered the translation along with its substrate toward the present position.

**Author Contributions:** Conceptualization, F.M., F.M.E., E.P. and M.P.; methodology, F.M., F.M.E., E.P. and M.P.; software, F.M.; validation, F.M, F.M.E., E.P. and M.P.; formal analysis, F.M and E.P.; investigation, F.M., F.M.E. and E.P.; resources, F.M., F.M.E., A.B. and M.P.; data curation, F.M, F.M.E, E.P., A.B. and M.P.; writing—original draft preparation, F.M.E., A.B. and M.P.; writing—review and editing, F.M., F.M.E., E.P., A.B. and M.P.; visualization, F.M.; supervision, F.M.E.; project administration, F.M.E.; funding acquisition, F.M.E., A.B. and M.P. All authors have read and agreed to the published version of the manuscript.

**Funding:** Funding was provided by FRA 2022—Università di Genova (resp. F. M. Elter, A. Briguglio, and M. Piazza).

**Data Availability Statement:** All the data are presented in the paper.

**Acknowledgments:** We would like to thank the reviewers for their critical comments that improved the quality of the original manuscript.

**Conflicts of Interest:** The authors declare no conflict of interest.

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
