# Peer review of "The Portofino Conglomerate (Eastern Liguria, Northern Italy): Provenance, Age and Geodynamic Implications"

_geosciences, doi:10.3390/geosciences13060154_

Round 1
Reviewer 1 Report (Previous Reviewer 3)
Dear All,
I have scrutinized the revised/resubmitted version of the manuscript: "The Portofino Conglomerate (Eastern Liguria, Northern Italy): provenance, age and geodynamic implications" of Mantovani et al.,
I have to congratulate with the authors, since they have nicely and convincingly addressed al my previous main comments.
I see the paper really improved, as so worth to be considered for publication on Geosciences.
I have only minor suggestions reported below:
1) please add in figure 1 a box pointing out the study area location. Please indicate also the location of units referred/discussed in the text (Voltri Group etc..)
2) in the captions of stereoplots (fig. 7, 9, 13) I suggest to rephrase in "equal area, lower hemisphere"
3) Line 329, please modify in " ultramilonitic levels characterized by a very fine grain size"
4) Line 377, please add the ")" after "poles."
5) Line 690, please modify in "Scholle, P. A."
I am not a native speaker, so I feel not qualified about this point. However, a minor check to typos would be great.
Author Response
Thanks for the comments and your much appreciated congratulations. In any case, we have further integrated the manuscript with your 5 annotations.
Best Regards
The authors
Reviewer 2 Report (Previous Reviewer 2)
The manuscript is further improved with respect to the past review when it was already good for publication. Hence, the manuscript deserves publication without further modification.
Author Response
Thanks for your positive comments.
Best Regards
The authors
Reviewer 3 Report (New Reviewer)
Dear Editor,
I read and commented the ms by Mantovani et al., “The Portofino Conglomerate (Eastern Liguria, Northern Italy): provenance, age and geodynamic implications.
I found it very interesting, although some work is needed before making it publishable.
There are many concepts that need to be simplified and/or placed in a table that could help the reader not keen with the Geology of the Northern Apennines.
I guess that the main point of the ms is that the Portofino Conglomerate deposited in a thrust top basin in the late Eocene-early Oligocene (or just Oligocene), surely the northern continuation of the wide basin that was the precursor of the Neogene Corsica Basin. On this, I suggest to the authors to read the paper by Cornamusini et al., 2002, and Cornamusini and Pascucci 2014 who dealt with similar items in southern zones.
Authors present a very detailed and convincing description of the Portofino Conglomerate that is the core of the ms, on the base of which the base the discussion and conclusions.
However, the good quality of data is shaded by the abuse of Apennine Unit names that made lost the poor reader. I have commented this many times in the attached pdf.
My suggestion is to simplify the ms using just names of Figure 1 and avoid entering as much as possible in complicate Geology of Norther Apennines.
Figure need a bit of revision making them more readable and avoiding duplicates.
I have done many, many comments and some English editing in the attached pdf.
On the base of the above consideration I suggest moderate to major revision and a bit of English revision as well as.
Last comment, the text I worked was not a final version but with not accepted corrections...so not easy to work with.
Sincerely,
Reviewer

English could be improved
Author Response
Dear reviewer,
Thank you for your helpful and constructive comments.
We are glad that the topic of the manuscript is of interest to you. We also thank you for the bibliographical indications you have suggested which have allowed us to better understand the geodynamic framework on which we have inserted some new considerations.
The strong correlation between our conclusions and those of the literature you suggest is evident. We think that a further collection of data and a further investigation of the basins of our areas could strengthen the correlations between all the tertiary basins in the geodynamic framework of the Western Mediterranean.
We accepted your corrections and implemented the manuscript with your suggestions. In particular, we have simplified some concepts making them easier to read; we have revised the English; all the names of the units are now mentioned on the basis of what is reported in figure 1, but we have left the specific names of the various units in brackets in order to be exhaustive also for those who know Ligurian geology.
However, we would like to answer a few questions you asked us in the PDF attachment:
1) Regarding the vertical faults, those shown in figure 2, only those classifiable through their fault surfaces and coeval kinematic indicators have been reported; these features are also discussed in the literature cited by us.
2) The contact between the substrate itself and the conglomerate is almost never visible both for geomorphological reasons and for the dense vegetation cover. In Figure 4 we have reported one of the few cases regarding the relationship between the Portofino Conglomerate and its substrate. The fractures are filled by clasts of the litho/petrofacies of Paraggi (basal). However, we have removed the term sin-sedimentary from the manuscript. We just wanted to point out that the tectonics involving the substrate (UA3) is not only post-depositional.
3) Figure 15: we understand his notation. However we preferred to produce two figures (Figure 15 and Figure 16); Figure 15 focuses only on the stratigraphic evolution of the PC in a synthetic and schematic way. The information reported is manifold (source lithology, nature of the clasts, litho/petrofacies differences) for which we preferred to exclude the regional and geodynamic representation. However, the regional and geodynamic representation is shown in Figure 16. We think that this facilitates the understanding of those non-expert readers who can thus focus their attention either on the stratigraphic aspect or on the geodynamic/regional one.
Best Regards
The authors
This manuscript is a resubmission of an earlier submission. The following is a list of the peer review reports and author responses from that submission.
Round 1
Reviewer 1 Report
The paper is very good. It can be improved with a better description of the data. They support fully the conclusions anyway.
Author Response
We thank you for the highly constructive criticisms that have allowed us to significantly improve the text. We have implemented all your annotations referring to both text and figures with the exception of the rose diagram and the table of sedimentological features. We think that the rose diagram would have added little to the scientific contribution of the paper given the different geographical position that the CP currently has compared to its initial position. We have implemented and improved the sedimentological descriptions and therefore we have deemed it unnecessary to insert the table. Best regards.
Reviewer 2 Report
This study describes the stratigraphic, petrographic, and paleontological features Portofino Conglomerate, a clastic wedge-top basin deposit in Eastern Liguria. The Authors provide a possible model of the clastic sources and the sedimentary formation of these deposits within the Alpine-Apennine orogenic evolution framework. In the following, I listed some suggestions and comments.
In the Abstract delete every acronym that you do not recall in the abstract.
Line 35: change auct. to Auct. and flysch to Flysch
Figure1: correct the thickness of Fmp in the geological cross-section to be consistent across the faults
Line 77: capitalize Formation when it is formalized. What is aka?
Line 96: always use PC for the Portofino Conglomerate
Lines 94-102: is this part necessary for this study?
Line 107: sometimes you use unit, other times Unit; please make this consistent in the whole paper.
Figure 2: the line is yellow, not white. Change folding structures to folds
Line 122: use HP-LT because you indicated before this acronym. Change Hogh to High
Line 128: add a reference for Ranzano sandstones and Tertiary Piedmont Basin.
Figure3: indicate UA3 in the picture and add a scale bar and picture orientation
Line 139: how can you state that there is no evidence of syn-depositional tectonic activity? Which activity? Syn-extension or syn-shortening?
Figure 4: add model after Vertical distribution. Change carbonatic to carbonate
Figure 5: add a scale bar and picture orientations. Change “fractures and white calcite veins. normal faults” to “fractures, white calcite veins and normal faults”
Line 205: ….and locally displaced…are they normal faults?
Line 210: can you insert a rose diagram of paleocurrent directions?
Line 227: change carbonate to calcareous
Figure 7: add a scale bar and picture orientations.
Line 255: can you insert a rose diagram of paleocurrent directions?
Figure8: add the scale bars
Figure 9: add a scale bar and picture orientations.
Figure10: add the scale bars
check “Figure” in bold in the whole text
line 324 and 389: change alpine to Alpine
line 345: somewhere, Genova is indicated as Genoa
line 351: change In fact to “,in fact”
line 354: change carbonatic to carbonate
line 356: change upper to Upper
line 401: (UA3)
line 402: delete “tectonic unit of”
line 409: delete chain
line 410: add a bracket at the end
line 415-423: this part has to move in the thin-section description. Not in the discussion
line 429-432: this part has to move in the thin-section description.
Line 437: indicate the “Apennine” vergence.
Line 438: delete “following, final”
Figure11: put in reverse order the pictures, “a” on the top
Strangely, the Portofino Basin formed following an extensional event, but the clastic deposits do not show syn-tectonic features.
Lines 494-497: this is a comment from a previous review?!? Delete
Author Response
We thank you for the highly constructive criticisms that have allowed us to significantly improve the text. We have implemented all your annotations referring to both text and figures except for the rose diagram and the annotation on the last figure (Figure 11). We think that the rose diagram would have added little to the scientific contribution of the paper given the different geographical position that the CP currently has compared to its initial position. As for the figure, we prefer to keep the reversed order of the letters so that it gives an idea of ​​the evolution of the basin, its uplift and its erosion.
Regarding the question on the syn-deposition tectonic activity we answer which can be easily observed only in one location (Porto Pidocchio) as the geomorphology of the promontory of Portofino leaves room for cliffs and rocky coast. Best regards.
Reviewer 3 Report
Editor Comment
Dear Editors, dear Authors, I have scrutinized the manuscript of Mantovani et al., "The Portofino Conglomerates (Eastern Liguria, Northern Italy): provenance, age and geodynamic implications" submitted to Geosciences.
The paper is focused on the geological description, including geological mapping, facies description and petrography of the clasts, of the Portofino Conglomerates (PC) in Eastern Liguria (Italy). This area is particular important to figure out the relationships between the Alps-Apennine orogenic systems. After their geological, petrographic and microstructural/paleontological observations, the authors recognized possible sources of the conglomerate clasts. Such sources are considered different units of the Alpine chain in the Western side of Liguria. The basin of PC is considered to be a piggy-back basin of the Alpine nappe pile, later on thrusted and deformed by Apennine tectonics. This Apennine overprint on former Alpine Units and top-thrust basins is widely documented in Liguria (see the recent paper of Schmid et al. 2017 SJG - ttps://doi.org/10.1007/s00015-016-0237-0 and references therein).
Unfortunately, my evaluation of the paper is, by far, not positive. The paper is not properly organized. It is difficult to read, and to understand the potential, to a not-specialized reader. Many units and places are introduced and discussed, without a clear logic. I am afraid that readers not familiar with this part of Italy and of Alpine-Apennine geology will have problems to follow this paper. My suggestion is to include a new figure 1, with a general geology of this part of Alpine-Apennine junction.
The structural geology description, and the overall structural and sedimentary framework, is quite confused and not well organized. There is a continuous mixing of observation/data and interpretation, when presenting the data, that should be always avoided in a scientific paper. Description of the metamorphic and igneous clasts, and their source attribution is, sincerely speaking, quite " naive".
Figure quality is often below the basics standards for a scientific journal, (just see figure one or note how scale bar are missing in the optical microscopy microphotos) etc... Many important references, pertinent with the topics of the paper, are missing (such as: Molli 2008; Molli et al., 2010; Schmid et al., 207; Elter & Pertusati, 1973, Sturani, 1973; Smye et al., 2021; Starr et al., 2020).
The English should be revised by a native speaker, since there are too many typos. Just check pag 4 line 122-124 or pag 15 line 494 - 497 (!!!).
Overall, I see this paper far to be mature enough to be considered for publication on an international peer-review journal, such as Geosciences (also in light of the recent disputes on MDPI/predatory list). So, I must recommend rejecting this paper. If the Authors will consider a new resubmission, the paper should be profoundly revised, also including my comments listed in the attached pdf.
General and Minor comments
I have done too many comments to be given in a report. I have highlighted them in the annotated attached PDF version of the paper.

Author Response
Dear revisor. We attached our answer to your revision.
Regards.

Round 2
Reviewer 2 Report
The Authors improved the paper following my suggestions; hence I accept it in its present form.
Best regards.
Author Response
Thank you so much for your important and helpful review.
Best regards
Reviewer 3 Report
It is a pity to see that most of my comments have been ignored.
I was not, or rather it was not my intention, to be sarcastic, as stated!
I made several key-comments, according to my point of view, aimed to improve the manuscript and to make the paper of a wider interest, also for people not specialist in the geology of Liguria and to improve the impact of the paper.
My opinion is still the one expressed previously.
Author Response
We are sorry that you have only considered the comments that we have not accepted, when a good part of them have been accepted and implemented in the text. The not accepted comments were justified by the attached word file in which we expressed our point of view and the reason for some of our choices.
Regards